Annales Geophysica, VOL. ???, XXXX, DOI:10.1029/,

# ₁ Transfer entropy and cumulant based cost as ₂ measures of nonlinear causal relationships in space ₃ plasmas: applications to $D_{st}$

Jay R. Johnson

₄ Andrews University, Berrien Springs, MI, USA

Simon Wing

₅ The Johns Hopkins University, Applied Physics Laboratory, Laurel,

₆ Maryland, USA

Enrico Camporeale

₇ Center for Mathematics and Computer Science (CWI), Amsterdam, The

₈ Netherlands

————

Jay R. Johnson, Andrews University, Berrien Springs, MI, 49104, USA. (jrj@andrews.edu)

Simon Wing, The Johns Hopkins University, Applied Physics Laboratory, Laurel, Maryland, 20723, USA (simon.wing@jhuapl.edu)

Enrico Camporeale, Center for Mathematics and Computer Science (CWI), Amsterdam, The Netherlands (e.camporeale@cwi.nl)

**Abstract.**     It is well known that the magnetospheric response to the so-

lar wind is nonlinear. Information theoretical tools such as mutual informa-

tion, transfer entropy, and cumulant based analysis are able to characterize

the nonlinearities in the system. Using cumulant based cost, we show that

nonlinear significance of $D_{st}$ peaks at $3-12$ hours lags that can be attributed

to $VBs$, which also exhibit similar behavior. However, the nonlinear signif-

icance that peaks at lags 25, 50, and 90 hours can be attributed to internal

dynamics, which may be related to the relaxation of the ring current. These

peaks are absent in the linear and nonlinear self-significance of $VBs$. Our

analysis with mutual information and transfer entropy show that both meth-

ods can establish that there are a strong correlation and transfer of infor-

mation from $V_{sw}$ to $D_{st}$ at a time scale that is consistent with that obtained

from the cumulant based analysis. However, mutual information also shows

that there is a strong correlation in the backward direction, from $D_{st}$ to $V_{sw}$,

which is counterintuitive. In contrast, transfer entropy shows that there is

no or little transfer of information from $D_{st}$ to $V_{sw}$, as expected because it

is the solar wind that drives the magnetosphere, not the other way around.

Our case study demonstrates that these information theoretical tools are quite

useful for space physics studies because these tools can uncover nonlinear

dynamics that cannot be seen with the traditional analyses and models that

assume linear relationships.

## 1. Introduction

One of the most practically important concepts in dynamical systems is the notion of causality. It is particularly useful to organize observational datasets according to causal relationships in order to identify variables that drive the dynamics. Understanding causal dependencies can also help to simplify descriptions of highly complex physical processes because it constrains the coupling functions between the dynamical variables. Analysis of those coupling functions can lead to simplification of the underlying physical processes that are most important for driving the system. It is particularly useful from a practical standpoint to understand causal dependencies in systems involving natural hazards because monitoring of causal variables is closely linked with warning.

A common method to establish causal dependencies in a data stream of two variables, e.g., $[a(t)]$ and $[b(t)]$, is to apply linear correlation studies such as *Strangeway et al.* [2005], which showed the relationship between downward Poynting flux and ion ouflows. Causal relationships are typically identified by considering a time-shifted correlation function

$$\lambda_{ab}(\tau) \triangleq \frac{\langle a(t)b(t+\tau)\rangle - \langle a\rangle\langle b\rangle}{\sqrt{\langle a^2\rangle - \langle a\rangle^2}\ \sqrt{\langle b^2\rangle - \langle b\rangle^2}} \tag{1}$$

where $\langle...\rangle$ is an ensemble average obtained by drawing samples at a set of measurement times, $\{t_0, t_1, ..., t_N\}$. For example, [*Borovsky et al.*, 1998] used such a method to identify relationships between solar wind variables and plasma sheet variables. The causal dependency that the plasma sheet responds to changes in the solar wind can be identified from the time-shift of the peak of the cross correlation indicating a response time. From this type of analysis it can be found that the plasma sheet generally responds from the

tail to the inner magnetosphere consistent with the notion of earthward convection. Such

analysis has been particularly useful to help understand plasma sheet transport.

However, the procedure of detecting causal relationships based on linear cross-correlation suffers from a number of limitations. First it should be noted that the statistical accuracy of the correlation function is limited by the resolution and length of the data stream. Second, the linear time series analysis ignores nonlinear correlations, which may be important for energy transfer in the magnetospheric system. For example, substorms are believed to involve storage and release of energy in the magnetotail, which is a highly nonlinear response. Similarly, magnetosphere-ionosphere coupling may also be highly nonlinear involving the nonlinear development of accelerating potentials along auroral field lines and nonlinear current-voltage relationships. Third, the cross-correlation may not be a particularly clear measure when there are multiple peaks or if there is little or no asymmetry in the forward [i.e., $\lambda_{ab}(\tau)$] and backward directions [i.e., $\lambda_{ba}(\tau) = \lambda_{ab}(-\tau)$]. Finally, the cross-correlation does not provide any way to clearly distinguish between two variables that are passively correlated because of a common driver rather than causally related.

In the remainder of this paper, we will discuss other methods to identify causal relationships based on entropy based discriminating statistics such as mutual information and transfer entropy. We will also discuss the cumulant-based method. We will illustrate the shortcomings and strengths of the various methods for studying causality with examples from nonlinear dynamics and space physics.

## 2. Linear vs Nonlinear Dependency

It is well known that the magnetosphere responds to variation in the solar wind param-
eters [*Clauer et al.*, 1981; *Baker et al.*, 1983; *Crooker and Gringauz*, 1993; *Papitashvili
et al.*, 2000; *Wing and Johnson*, 2015; *Johnson and Wing*, 2015; *Wing et al.*, 2016], and
it has been established that the magnetosphere has a significant linear response to the
solar wind. However, it is also expected that the magnetosphere has a nonlinear response
[*Tsurutani et al.*, 1990; *Vassiliadis et al.*, 1990; *Klimas et al.*, 1998; *Valdivia et al.*, 2013;
*Balikhin et al.*, 2011]. The nonlinear response may driven by internal dynamics rather
than driven externally [*Wing et al.*, 2005; *Johnson and Wing*, 2005]. For example, the
internal dynamics associated with loading and unloading of magnetic energy associated
with storms and substorms is nonlinear [e.g., *Johnson and Wing*, 2014, and references
therein]. Indeed, the data analysis of *Bargatze et al.* [1985] indicated that the dynamical
response of the magnetosphere to solar wind input could not be entirely understood using
linear prediction filters.

Suppose that we consider a set of variables $\mathbf{a}$ and $\mathbf{b}$ which could be vectors of variables
measured in time and we would like to measure their dependency. Instead of consider-
ing the covariance matrix/correlation function, we consider a more general measure of
dependency between an input and output is obtained by considering whether

$$P(\mathbf{a}, \mathbf{b}) \overset{?}{=} P(\mathbf{a})\mathbf{P}(\mathbf{b}). \tag{2}$$

where $P(\mathbf{a}, \mathbf{b})$ is the joint probability of input $\mathbf{a}$ and output $\mathbf{b}$ while $P(\mathbf{a})$ and $P(\mathbf{b})$ are
the probability of $\mathbf{a}$ and $\mathbf{b}$ respectively. If the relationship holds, then the variables $\mathbf{a}$
and $\mathbf{b}$ are independent. For all other cases, there is some measure of dependency. In the
case where the system output is completely known given the input, $P(\mathbf{a}, \mathbf{b}) = \mathbf{P}(\mathbf{a})$. The

83 advantage of considering Equation 2 is that it is possible to detect the presence of higher

order nonlinear dependencies between the input and output even in the absence of linear

dependencies [*Gershenfeld*, 1998].

## 2.1. Mutual Information and Cumulant based cost

Mutual information and cumulant-based cost are two useful measures that quantify

Eq. 2. Mutual information has the advantage that in the limit of Gaussian joint proba-

bility distributions, it may be simply related to the correlation coefficient $C_{ab}(\tau)$ defined

in equation 1 [*Li*, 1990]. Cumulants have the advantage of good statistics for limited

datasets and noisy systems [*Deco and Schürmann*, 2000]. Moreover, for high-dimensional

systems it is more efficient to compute moments of the data rather than try to construct

the probability density function.

Correlation studies also only detect linear correlations, so if the feedback involves non-

linear processes (highly likely in this case) then their usefulness may be seriously limited.

Alternatively, entropy-based measures such as mutual information [*Prichard and Theiler*,

1995; *Materassi et al.*, 2011] and cumulants [*Johnson and Wing*, 2005] are useful for de-

tecting linear as well as nonlinear correlations. The mutual information is constructed

from the probability distribution function of the variables and may be computed using

an quantization procedure where data is binned such that the samples $[a(t)]$ are assigned

discrete values $\hat{a} \in \{a_1, a_2, ..., a_n\}$ of an alphabet $\aleph_1$ and $[b(t)]$ is assigned discrete values

$\hat{b} \in \{b_1, b_2, ..., b_m\}$ of an alphabet $\aleph_2$. The *ad hoc* time-shifted mutual entropy

$$\mathcal{M}_{ab}(\tau) \triangleq \sum_{\hat{a} \in \aleph_1, \hat{b} \in \aleph_2} p(\hat{a}(t+\tau), \hat{b}(t)) \log \left( \frac{p(\hat{a}(t+\tau), \hat{b}(t))}{p(\hat{a})p(\hat{b})} \right) \tag{3}$$

has been used as an indicator of causality, but suffers from the same problems as time-

shifted cross correlation when it has multiple peaks and long range correlations.

Similarly, examination of time-shifted cumulants could be used as an indicator of causal-

ity in a nonlinear system. In this case, we can define a discriminating statistic

$$D^C = \sum_{q=1}^{\infty} \sum_{i_1,\ldots,i_q \in \Pi_q} K^2_{1i_2\ldots i_q} \tag{4}$$

where

$$
\begin{aligned}
K_i &= C_i = \langle z_i \rangle \tag{4}\\
K_{ij} &= C_{ij} - C_i C_j = \langle z_i z_j \rangle - \langle z_i \rangle \langle z_j \rangle\\
K_{ijk} &= C_{ijk} - C_{ij} C_k - C_{jk} C_i - C_{ik} C_j + 2C_i C_j C_k\\
K_{ijkl} &= C_{ijkl} - C_{ijk} C_l - C_{ijl} C_k - C_{ilk} C_j - C_{ljk} C_i\\
&\quad - C_{ij} C_{kl} - C_{il} C_{kj} - C_{ik} C_{jl} + 2(C_{ij} C_k C_l\\
&\quad + C_{ik} C_j C_l + C_{il} C_j C_k + C_{jk} C_i C_l + C_{jl} C_i C_k\\
&\quad + C_{kl} C_i C_j) - 6C_i C_j C_k C_l
\end{aligned}
$$

are the cumulants

$$C_{i\ldots j} = \int d\mathbf{z} P(\mathbf{z}) z_i \ldots z_j \equiv \langle z_i \ldots z_j \rangle \tag{5}$$

of the joint probability distribution for variables $z_1, \ldots, z_j$.

With only two variables, $a$ and $b$, defined above, we can consider the cost function

$$D^C_{a,b}(\tau) = D_C(a(t), b(t+\tau)) \tag{6}$$

The presence of nonlinear dependence has been identified by comparing the cumulant cost

for a time series with the cumulant based cost of surrogate time series, which are con-

structed to have the same linear correlations as in [*Johnson and Wing*, 2005]). Significance

measures the difference in the discriminating statistic from the mean of the discriminating

statistic of the surrogates in terms of the spread of the surrogates, $\sigma$.

In Section 3, we will show an application of cumulant based analysis to the distur-

bance storm-time index ($D_{st}$). In principle, the cross-correlation, mutual information,

and cumulant-based cost should be independent of the selection of measurement points

if the system is stationary; therefore, time stationarity can be examined by comparing

these discriminating statistics for groups of measurements drawn from different windows

of time as in [*Johnson and Wing*, 2005; *Wing et al.*, 2016].

## 2.2. Transfer entropy

Another method for determining causality is the one-sided transfer entropy [*Schreiber*,

2000; *De Michelis et al.*, 2011; *Materassi et al.*, 2014; *Wing et al.*, 2016, 2018], which is

based upon the conditional mutual information

$$\mathcal{M}_C(x,y|z) \triangleq \sum_{x \in \aleph_1} \sum_{y \in \aleph_2} \sum_{z \in \aleph_3} p(x,y,z) \log\left(\frac{p(x,y,z)p(z)}{p(x,z)p(y,z)}\right) \tag{7}$$

The conditional mutual information measures the dependence of two variables, $x$ and $y$,

given a conditioner variable, $z$. If either $x$ or $y$ are dependent on $z$ the mutual information

between $x$ and $y$ is reduced, and this reduction of information provides a method to

eliminate coincidental dependence, or conversely to identify causal dependence.

Transfer entropy considers the conditional mutual information between two variables

using the past history of one of the variables as the conditioner.

$$\mathcal{T}_{a \to b}(\tau) = \sum_{\hat{a} \in \aleph_1} \sum_{\hat{a}^{(k)} \in \aleph_1^{(k)}} \sum_{\hat{b} \in \aleph_2} p(\hat{a}(t+\tau), \hat{a}^{(k)}(t), \hat{b}(t)) \log\left(\frac{p(\hat{a}(t+\tau)|\hat{a}^{(k)}(t), \hat{b}(t))}{p(\hat{a}(t+\tau)|\hat{a}^{(k)}(t))}\right) \tag{8}$$

where $\hat{a}^{(k)}(t) = [\hat{a}(t), \hat{a}(t-\Delta), ..., \hat{a}(t-(k-1)\Delta)]$. The standard definition of transfer

entropy takes $k = 1$ (no lag), but keeping a higher embedding dimension could in prin-

ciple provide a more precise measure (for example, if $a$ has periodicity a dimension of 2

may provide better prediction of future values of $a$ from its past time series and therefore

lower the transfer entropy. Transfer entropy as a discriminating statistic has the following

advantages. First in the absence of information flow from $a$ to $b$ (i.e., $a(t+\tau)$ has no

additional dependence from $b(t)$ beyond what is known from the past history of $a^{(k)}(t)$)

$p(\hat{a}(t+\tau)|\hat{a}^{(k)}(t), \hat{b}(t)) = p(\hat{a}(t+\tau|\hat{a}_{(k)}(t))$ and the transfer entropy vanishes. The transfer

entropy is also highly directional so that $\mathcal{T}_{a \to b} \neq \mathcal{T}_{b \to a}$. The advantage can be clearly

seen for dynamical systems where variables are forward differenced and the transfer en-

tropy is clearly one-sided while mutual information and correlation functions can even be

symmetric [*Schreiber*, 2000]. This measure also accounts for static internal correlations,

which can be used to determine whether two variables are driven by a common driver or

whether the variable $b$ is causally driving the variable $a$.

Both mutual information and transfer entropy require binning of data. As mentioned

in *Wing et al.* [2016], the number of bins $(n_b)$ needs to be chosen properly and there are

some guidelines that can be followed. In general, we would like to maximize the amount

of information. Having too few bins would lump too many points into the same bin,

leading to loss of information. Conversely, having too many bins would leave many bins

with 0 or a few number of points, which also would lead to loss of information. *Sturges*

[1926] proposed that for a normal distribution, optimal $n_b = log_2(n) + 1$ and bin width

$w = range/n_b$, where $n$ = number of points in the dataset, $range$ = maximum value $-$

minimum value of the points. In practice, there is usually a range of $n_b$ that would work.

## 3. Application to space weather: $D_{st}$ analysis

$D_{st}$ (disturbance storm time index) is an hourly index that gives a measure of the

strength of the symmetric ring current that, in turn, provides a measure of the dynamics

of geomagnetic storms [*Dessler and Parker*, 1959]. Because of its global nature, $D_{st}$ is

often used as one of the several indices that represent the state of the magnetosphere.

For example, *Balasis et al.* [2011] used the cumulative square amplitude of $D_{st}$ time series

as a proxy for energy dissipation rate in the magnetosphere and found that it fits well

a power law with log-periodic oscillations, which was interpreted as evidence for discrete

scale invariance in the $D_{st}$ dynamics.

When plasma sheet ions are injected into the Earth inner magnetosphere, they drift

westward around the Earth, forming the ring current. Studies have shown that the

substorm occurrence rate increases with solar wind velocity (high speed streams) [e.g.,

*Kissinger et al.*, 2011; *Newell et al.*, 2016]. An increase in the solar wind electric field,

$VB_z$, can increase the dawn-dusk electric field in the magnetotail, which in turn deter-

mines the amount of plasma sheet particles that move to the inner magnetosphere [e.g.,

*Friedel et al.*, 2001]. Studies have shown that the electric field, $VBs$ ($V_{sw}$ × southward

IMF $B_z$) or $VB_z$, has a strong effect on the ring current dynamics [*Burton et al.*, 1975;

*O'Brien and McPherron*, 2000; *McPherron and O'Brien*, 2001; *Weygand and McPherron*,

2006].

For the present study, we examine the relationships between solar wind velocity ($V_{sw}$)

and $VBs$ with $D_{st}$. We use $D_{st}$ records in the period $1974 - 2001$ obtained from

Kyoto University World Data Center for Geomagnetism (http://swdcwww.kugi.kyoto-

u.ac.jp/index.html). The corresponding solar wind data are obtained from IMP-8, ACE,

WIND, ISEE1, and ISEE3 observations. The ACE SWEPAM and MAG data; and

the WIND MAG data are obtained from CDAWeb (http://cdaweb.gsfc.nasa.gov/). The

WIND 3DP data are obtained from the 3DP team directly. The ISEE1 and ISEE3

data are obtained from UCLA (these datasets are also available at NASA NSSDC

[http://nssdc.gsfc.nasa.gov/space/]). The IMP8 data come directly from the IMP teams.

The solar wind is propagated with minimum variance technique [*Weimer et al.*, 2003] to

GSM (X, Y, Z) = (17, 0, 0) $R_E$ to produce 1-min files, from which hourly averaged solar

wind parameters are constructed.

### 3.1. Cumulant based analysis

Section 2.1 presents the method of cumulant based cost. Here, we show an application

of cumulant based cost to detect nonlinear dynamics in $D_{st}$. We consider the forward

coupling between a solar wind variable such as $VBs$ and $D_{st}$, which characterizes the

ring current response to the solar wind driver. We therefore consider the nonlinear cross-

correlations of the vector

$$\mathbf{c}(t, \tau) = \{VBs(t), D_{st}(t + \tau)\} = \{z_1, z_2\} \tag{9}$$

The generalization of cost is based on realizations of $\{z_1, z_2\}$. In this case, each variable

is Gaussianized with unit variance to eliminate static nonlinearities (i.e. higher order

self-correlations in $VB_s$ and $D_{st}$ are eliminated so that the cost measures only cross-

dependence between $VBs$ and $D_{st}$). This procedure is explained in the next paragraph.

The distribution of $D_{st}$ and $VBs$ are generally non-Gaussian. As such, the raw dis-

tributions (e.g., distribution of values of $D_{st}$) may have nonzero higher-order cumulants

(e.g., they can have a skew and kurtosis). This property makes it more difficult to in-

terpret whether the higher order cumulants in the time evolution arise from the overall

shape of the distribution of data points or from the time-ordering of the data. To elim-

inate the inherent nonzero cumulants in the overall distribution of data, we construct a

rank-ordered map from the original dataset to a proxy dataset of the same length drawn

from a Gaussian distribution [*Kennel and Isabelle*, 1992; *Schreiber and Schmitz*, 1996;

*Deco and Schürmann*, 2000]. The distribution of the proxy dataset ensures that all cu-

mulants of the distribution beyond second order should in principle vanish. However, the

time-ordering of the data can still lead to nonzero cumulants, because the joint probability

distribution of $D_{st}(t+\tau)$ and $D_{st}(t)$ may be non-Gaussian even if the distribution of $D_{st}$ is

Gaussian. Moreover, it is simple to construct surrogate data from the Gaussianized data

that shares the same autocorrelation by using the same power spectrum, but randomly

shifting the phases of the Fourier coefficients. The surrogate data therefore has the same

autocorrelation as the original data. Any deviation from the linear statistic is apparent

from comparison with the surrogate data, and we interpret these deviations as evidence

of nonlinear dependence because we have falsified the hypothesis that the data can be

adequately described by linear statistics. This method has been successfully employed in

*Johnson and Wing* [2005] where $K_p$ record was analyzed with mutual information and

cumulants.

In Figure 1 we plot the significance obtained from the year 1999 as a function of time

delay, $\tau$. Significance extracted from $\{VBs(t), D_{st}(t + \tau)\}$ and $\{VBs(t), VBs(t + \tau)\}$

for 1999 are plotted in panels (a) and (b), respectively. It should be noted that there

is a strong linear response at around 3 hour time delay. As shown in Figure 1a, there

is a clear nonlinear response with peaking around $3-10$, 25, 50 and 90 hours lasting for

approximately 1 week. In contrast, in Figure 1b, the nonlinearity only has one broad peak

around $3 - 12$ hours in the self-significance for $VBs$, suggesting that the nonlinear and

linear peaks at $\tau = 3-12$ hours in in Figure 1a i may be associated with $VBs$. We will

revisit the solar wind causal relationship with $D_{st}$ using transfer entropy in Section 3.2.

The absence of the nonlinear peaks at $\tau = 25$, 50, and 90 hours in the self-significance

for $VBs$ (Figure 1b) suggest that these nonlinearities in $\{VBs(t), D_{st}(t+\tau)\}$ are related to

internal magnetospheric dynamics. As the $D_{st}$ index is thought to reflect storm activity,

it is reasonable that nonlinear significance would decay on the order of 1 week as storms

commonly last around that time. The strong nonlinear responses at $\tau = 25$, 50, and 90

203 hours are likely related to multiple modes of relaxation of the ring current following the

204 commencement of storms. It should also be noted that other nonlinearities detected by

205 even higher order cumulants may also be present; however, the calculation demonstrates

the nonlinear nature of the underlying dynamics.

A common scenario for storm-ring current interaction is the following. A storm com-

presses the magnetosphere, intensifies the magnetic field in the magnetosphere, and injects

energetic particles into the ring current region. The ring current intensifies during the

main phase of the storm, which can last $\sim 6$ hours [*Weygand and McPherron*, 2006].

Once the injection stops, the ring current begins to decay and the storm enters the re-

covery phase. Conservation of magnetic moment implies that anisotropies develop in the

ring current and plasma sheet. Anisotropy drives the ring current plasma unstable to ion

cyclotron waves. The ion cyclotron waves scatter energetic ions into the loss cone so that

they are lost from the ring current. Nonlinear interaction between waves and particles

keeps the plasma near marginal instability with a steady loss of energetic particles due

to wave-particle scattering. Other loss mechanisms include charge exchange, coulomb

scattering, and convective of ions to the front of the magnetopause. The ring current

decay can have two stages [*Kozyra et al.*, 2002]. In the first stage, the ring current decays

rapidly and the loss mechanisms can be attributed to convective out flow, pitch-angle

scattering in the ring current, and $O^+$ charge exchange [e.g., *Weygand and McPherron*,

2006; *Hamilton et al.*, 1988]. The second stage may typically begin about one day from

the commencement of the storm (see, for example, Figure 7 of *Kozyra et al.* [2002]). In
the second stage, the decay rate is slower and is attributed mainly to $H^+$ charge exchange
[*Hamilton et al.*, 1988] and can take several days to deplete the ring current to the baseline
level [*Smith et al.*, 1976]. We can speculate that the multiple nonlinear response lag times
that are detected with the cumulant-based approach are likely the relaxation of the ring
current due to complex interplay of multiple loss processes.

### 3.2. Transfer entropy

As mentioned in Section 2.2, transfer entropy gives a measure of how much information
is transfered from one variable to another. We have applied transfer entropy and mutual
information to the relationship between the $V_{sw}$ and $D_{st}$ for the period $1974 - 2001$. The
result is shown in Figure 2. Note that the mutual information measure suggests strong
correlations between prior values of $D_{st}$ and $V_{sw}$. This finding suggests that $D_{st}$ could be
a driver of $V_{sw}$, which is counterintuitive. On the other hand, the transfer entropy clearly
shows that this information transfer in the backward direction $(D_{st} \rightarrow V_{sw})$ does not rise
above the noise level (the horizontal blue lines indicate mean and standard deviation of
100 surrogate data sets where the data was randomly reordered.) This result is expected
because it is the solar wind that drives the magnetosphere, not the other way around.
The transfer of information from $V_{sw}$ to $D_{st}$ peaks at $\tau = 8 - 11$ hours. The cumulant
based analysis in Section 3.1 shows that the response of $D_{st}$ to $VBs$ has similar time scale.
This time scale is consistent with the 4 to 15 hours transport time for the solar wind to
reach the midnight and noon regions of the geosynchronous orbit, respectively, from the
dayside magnetopause [*Borovsky et al.*, 1998]. The analysis presented here illustrates the
power of the transfer entropy for accessing causality.

## 4. Summary

We recently used mutual information, transfer entropy, and conditional information to discover the solar wind drivers of the outer radiation belt electrons [*Wing et al.*, 2016]. Because $V_{sw}$ anticorrelates with solar wind density ($n_{sw}$), it is hard to isolate the effects of $V_{sw}$ on radiation belt electrons, given $n_{sw}$ and vice versa. However, using conditional mutual information, we were able to determine the information transfer from $n_{sw}$ or any other solar wind parameters to radiation belt electrons, given $V_{sw}$ (or any other solar wind parameters). We also showed that the triangle distribution in the radiation belt electron vs. solar wind velocity plot [*Reeves et al.*, 2011] can be understood better when we consider that $V_{sw}$ and $n_{sw}$ transfer information to radiation belt electrons with 2 days and 0 day ($< 24$ hr) lags, respectively. Also recently, we used transfer entropy to better understand the causal parameters in the solar cycle and their response lag times [*Wing et al.*, 2018].

As a follow up to *Wing et al.* [2016, 2018], the present study demonstrates further how information theoretical tools can be useful for space physics and space weather studies. Cumulant based analysis can be used to distinguish internal vs. external driving of the system. Both mutual information and transfer entropy give a measure of shared information between two variables (or vectors). However, unlike mutual information, transfer entropy is highly directional. To illustrate, we apply mutual information, transfer entropy, and cumulant based analysis to investigate the dynamics of $D_{st}$ index.

Our analysis with mutual information and transfer entropy indicates that there are strong linear and nonlinear correlations and transfer of information, respectively, in the forward direction between $V_{sw}$ and $D_{st}$ ($V_{sw} \rightarrow D_{st}$). However, mutual information indi-

cates that there is also a strong correlation in the backward direction ($D_{st} \to V_{sw}$), which

is puzzling and counterintuitive. In contrast, the transfer entropy indicates that there is

no information transfer in the backward direction ($D_{st} \to V_{sw}$), as expected because it is

the solar wind that drives the magnetosphere, not the other way around. The transfer of

information from $V_{sw}$ to $D_{st}$ peaks at $\tau = 8 - 11$ hours.

Using the cumulant-based significance, we have established that the underlying dynam-

ics of $D_{st}$ is in general nonlinear exhibiting a quasiperiodicity which is detectable only if

nonlinear correlations are taken into account. The strong nonlinear responses of $D_{st}$ to

$VBs$ at $\tau = 25$, 50, and 90 hours are likely related to multiple modes of relaxation of the

ring current from multiple loss mechanisms following the commencement of storms. It is,

of course, possible that these nonlinearities are caused by solar wind drivers other than

$VBs$. However, the timing of these nonlinearities would put them well in the recovery

phase of a storm and previous studies suggested that the ring current decays in the recov-

ery phase are strongly influenced by $VBs$ [*Burton et al.*, 1975; *O'Brien and McPherron*,

2000; *McPherron and O'Brien*, 2001]. The nonlinearities at $\tau = 3 - 12$ hours are not

caused by internal dynamics but rather by the solar wind driver, which is similar with

the time scale for the solar wind transport time from the dayside magnetopause to the

inner magnetosphere. This time scale is consistent with the time scale for the information

transfer from the solar wind to $D_{st}$ obtained from transfer entropy analysis.

Although linear models are useful, our results indicate that these models have to be

used with cautions because solar wind $-$ magnetosphere system is inherently nonlinear.

Hence, nonlinearities generally need to be taken into account in order to describe the

system accurately. Local-linear models (which include slow evolution of parameters) may

be able to handle some nonlinearities, but it is expected that these local-linear models

would have difficulties if the dynamics suddenly and rapidly change.

**Acknowledgments.** All the derived data products in this paper are available

upon request by email (simon.wing@jhuapl.edu).   Simon Wing acknowledges sup-

port from JHU/APL Janney Fellowship, NSF Grant AGS-1058456, and NASA Grants

(NNX13AE12G, NNX15AJ01G, NNX16AR10G, and NNX16AQ87G). Jay R. Johnson ac-

knowledges support from NASA Grants (NNH11AR07I, NNX14AM27G, NNH14AY20I,

NNX16AC39G), NSF Grants (ATM0902730, AGS-1203299, AGS-1405225), and DOE

contract DE-AC02-09CH11466.   E. Camporeale is partially funded by the NWO-Vidi

grant No. 639.072.716. We thank James M. Weygand for the solar wind data processing.

The raw solar wind data from ACE, Wind, ISEE1 and ISEE3 were obtained from NASA

CDAW and NSSDC.

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

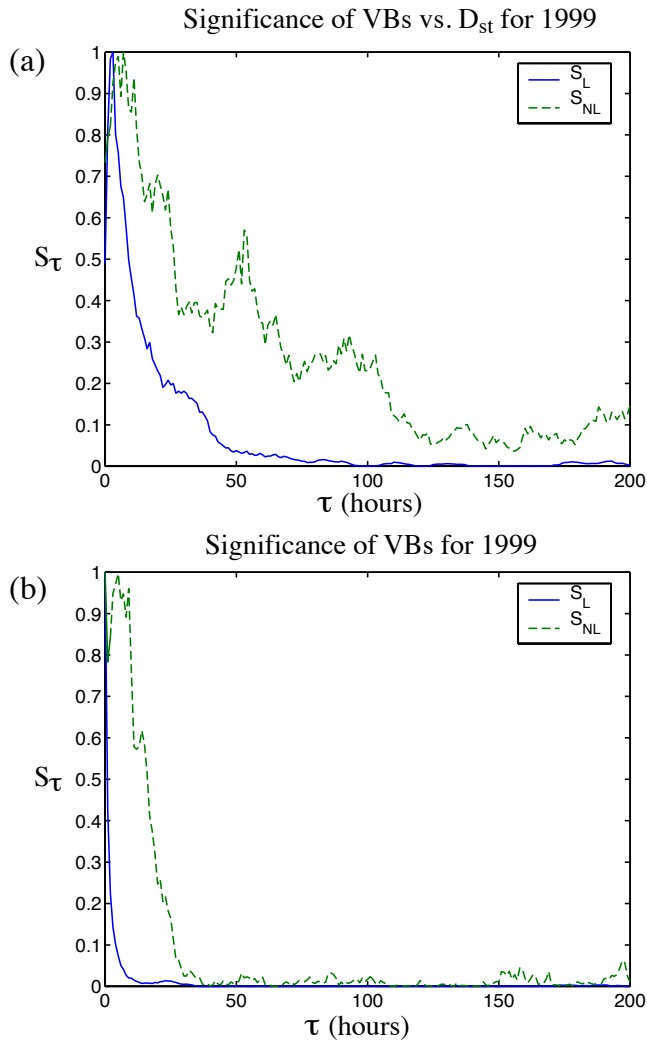

**Figure 1.** Significance extracted from (a) $\{VBs(t), D_{st}(t-\tau)\}$ and (b) $\{VBs(t), VBs(t-\tau)\}$ for 1999. It should be noted that there is a strong linear response at around 3 hour time delay. There is a clear nonlinear response with a strong peak around 50 hours lasting for approximately 1 week. The longterm nonlinear response is absent in the solar wind data indicating that the longterm nonlinear correlations between $VBs$ and $D_{st}$ are the result of internal magnetospheric dynamics.

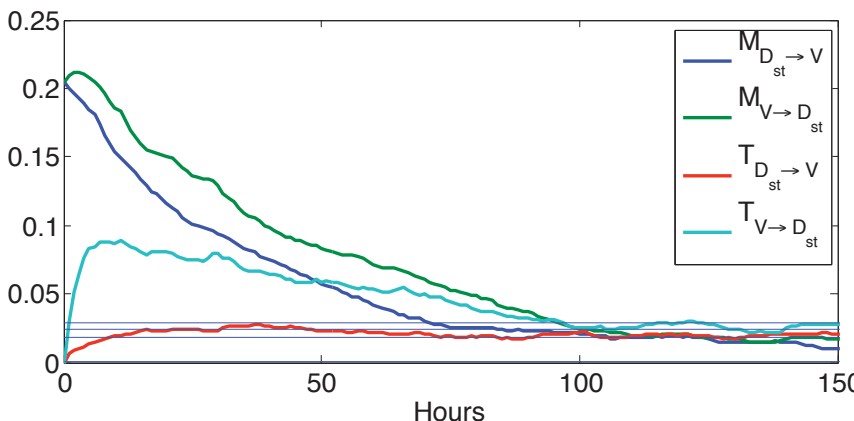

**Figure 2.** Comparison of mutual information and transfer entropy measures to determine causal driving of the magnetosphere as characterized by $D_{st}$. Note that causal driving appears to peak somewhat later (11 hours) than indicated by mutual information (2 hours) indicating that internal dynamics likely are very important initially. The backward transfer entropy is below the noise level for all values indicating that $D_{st}$ in no way influences the upstream solar wind velocity. Such a conclusion could not be inferred from the mutual information measure.