# Peer review of "Transfer entropy and cumulant based cost as measures of nonlinear causal relationships in space plasmas: applications to $D_{st}$"

_Annales Geophysicae, 2018_

## Referee Comment (RC2) · Anonymous Referee #2 · 28 Mar 2018

Report on Ms. angeo-2018-7 (Johnson et al.)
"Transfer entropy and cumulant based cost as measures of nonlinear causal relationships in space plasmas: applications to Dst"

In this work the Authors shows an application of some statistical and information theory-based methods to the study of the Earth's magnetosphere response to solar wind changes with the aim to demonstrate that these methods and tools can be useful to study the nonlinear dynamics of the Earth's magnetosphere. In particular, the Authors applied two quite novel methods, the transfer entropy analysis and the cumulant based cost, to the investigation of the causal relationship between VBs, Vsw and Dst, showing how some of the information contained in the dynamical evolution of Dst are not directly to solar wind driving. The manuscript is quite well written. However, although I believe that the topic treated in this work is timely appropriate and the techniques presented could be of interest of a wide community as the one of Annales Geophysicae, I think that there are some aspect and points of this work that need to be revised before considering it ready for publication.

Thus, I recommend to send back the manuscript to the Authors for a major revision according to the points listed below.

Major Questions.

1) In the overall paper (Introduction, Linear vs Nonlinear Dependency, etc.) the Authors miss to cite several previous topical works dealing with the nonlinear and complex dynamics of the Earth's magnetosphere (e.g. Tsurutani, B., et al., GRL, 1990; Vassiliadis, et al., GRL, 1990. Klimas, et al., JGR, 1996; etc.). The same is for what regards previous application of information theory methods to space plasma physics and the Earth's magnetosphere. I strongly invite the Authors to revise their introduction and manuscript considering more extensively the previous literature.

2) I would like to understand why the Authors in making their analysis do not consider instead of Dst its high-resolution version, Sym-H. Indeed, in disentangling the internal magnetospheric dynamics with respect to the external driven one the use of Dst index could be not sufficient, because all the fast internal processes are not contained in this index. I would like to stress that the internal magnetospheric dynamics is generally related to processes taking place in the tail regions which are characterised by timescales shorter than 60-90 minutes. Thus, Dst cannot be able to provide a reasonable information on it. Please, comment your choice and justify it.

3) In section 3.1, Cumulant based analysis, the Authors state that each of the considered variables is Gaussianized. I do not understand this statement. The PDFs of Dst and also external drivers is generally not Gaussian. What do they mean with this statement ? I guess that probably they refer to the fact that time series are normalized to unit variance. Please explain better this statement.

4) If I have correctly understood the cumulant based method, the nonlinear cross-correlation quantity should provide an information of the overall (linear and nonlinear) correlation between VBs and Dst. Thus, how can the Authors state that peaks at 25, 50 and 90 hours are of an internal origin on the basis that they are not present in the auto-correlation of external drivers ? Furthermore, in doing their analysis the Authors have considered Dst records covering 27 years (1974-2001) without discriminating between single geomagnetic storms and multiple geomagnetic storms. So how they can assert that these secondary peaks (which is less prominent) do not come from such multiple geomagnetic storms but reflects internal processes ? This conclusion seems to me not convincing. To convince the reader that there are secondary peaks in the nonlinear cross-correlation that are of an internal origin, the Authors should make the analysis on a subset of geomagnetic storms which are characterised by only a single negative-peak in Dst.

5) Page 11. To my knowledge there should be also other processes/mechanisms than ion cyclotron waves -particle scattering that could be responsible for ring-current decay. For instance, I remember that also ENA loss mechanisms could contribute to the decay of the ring current. Perhaps, this could be considered in discussing this point.

6) In the Transfer Entropy analysis section few details are given about the way Transfer Entropy and Mutual Information are computed. To my knowledge binning procedure and PDF computing method are critical issues in evaluating these quantities. I believe that more information should be provided to make the reader able to reproduce the results.

7) The result on the time delay (8-11 hr) between the information transfer from Vsw and Dst looks very long. The Earth's magnetosphere is expected to respond to solar wind changes on shorter timescale and this is also the case of ring-current. This is also corroborated by the capability of several Artificial Neural Network models of the Earth's magnetospheric response that consider a time delay of 1-2 hours as input variables for predicting Dst (see e.g. Wu and Lundstedt, JGR,

1997; Lundstedt et al., GRL, 2002; Pallocchia et al., Ann. Geophys., 2006). The Authors should motivate this result with more physical considerations.

8) Figure 1 is hardly readable. I suggest to expand the X-axis or to include a inset where the first part of X-axis is expanded.

Minor points.

Some references are missing (there are some question marks at page 8.

---

## Author Comment (AC1) · 20 Apr 2018

The comment was uploaded in the form of a supplement:
https://www.ann-geophys-discuss.net/angeo-2018-7/angeo-2018-7-AC1-
supplement.zip

---

## Author Comment (AC3) · 25 Apr 2018

**Response to reviewer**

We thank the reviewer for the helpful comments. We have revised the manuscript accordingly. Our point by point response to the review is given below.

Major Questions.

*1) In the overall paper (Introduction, Linear vs Nonlinear Dependency, etc.) the Authors miss to cite several previous topical works dealing with the nonlinear and complex dynamics of the Earth's magnetosphere (e.g. Tsurutani, B., et al., GRL, 1990; Vassiliadis, et al., GRL, 1990. Klimas, et al., JGR, 1996; etc.). The same is for what regards previous application of information theory methods to space plasma physics and the Earth's magnetosphere. I strongly invite the Authors to revise their introduction and manuscript considering more extensively the previous literature.*

We have added the references in Sections 2, 2.1, 2.2 (Linear vs nonlinear, mutual information and cumulant based cost, transfer entropy). We intend this paper to be a short paper that fits Annales Geophysica Communicate format (a few pages long). This constrains the number of citations not just in the introduction, but throughout the entire paper. Nonetheless, we added the references that the reviewer suggested as well as some additional references to provide additional background.

*2) I would like to understand why the Authors in making their analysis do not consider instead of Dst its high-resolution version, Sym-H. Indeed, in disentangling the internal magnetospheric dynamics with respect to the external driven one the use of Dst index could be not sufficient, because all the fast internal processes are not contained in this index. I would like to stress that the internal magnetospheric dynamics is generally related to processes taking place in the tail regions which are characterised by timescales shorter than 60-90 minutes. Thus, Dst cannot be able to provide a reasonable information on it. Please, comment your choice and justify it.*

We agree with the referee that there are fast internal processes in the tail that are not contained in the Dst index. This limitation would be particularly relevant if we were studying onset phemonena or the initiation of fast flows. For such phenomena it would be much better to use AE or PC or other indices that have high time resolution. Although fast processes in the tail may be relevant to the storm or substorm initiations, which would start the process of particle injections into the magnetosphere, the processes that govern the ring current dynamics are completely different than those that initiate the storm or substorm onsets in the tail. We refer the referee to our previous studies that addressed onset phenomena with these methods on much shorter timescales (1 minute resolution) [Johnson and Wing, External vs Internal Triggering of Substorms: An Information-Theoretical Approach, 2014]. These type of initiation studies are not within the scope of the present study.

For the work described in the paper, we are interested in the dynamics of the Dst and the symmetric ring current (Dst has long been used as a proxy for the symmetric ring current [Rostoker, 2000]). It is well known that the ring current takes a long time build up and decay. For example, Weygand and McPherron [2006] found that the ring current growth time is about 6 hours and decay time > 72 hours. The same study attributed the growth phase to the driving

of the solar wind electric field. Several processes have been identified as the causal agents for the ring current decays such as convections of ions out of the front of the magnetosphere, scattering into the loss cone due to wave-particle interactions, ENA charge exchange etc. All these processes reduce the ring current slowly, in the order of ten hours to a few days [e.g., Feldstein et al., 1990; MacMahon and Llop-Romero, 2008]. Hence, the Dst index is adequate for the study that we presently pursue. We have expanded a discussion on the long time scale of Dst dynamics in Section 3.1

3) In section 3.1, Cumulant based analysis, the Authors state that each of the considered variables is Gaussianized. I do not understand this statement. The PDFs of Dst and also external drivers is generally not Gaussian. What do they mean with this statement ? I guess that probably they refer to the fact that time series are normalized to unit variance. Please explain better this statement.

The distribution of Dst and VBs are indeed generally nongaussian. As such, the raw distributions (e.g. distribution of values of Dst) have nonzero higher-order cumulants (e.g., they can have a skew and kurtosis). This property makes it more difficult to interpret whether higher order cumulants in the time evolution arise from the overall shape of the distribution of datapoints or from the time-ordering of the data. To eliminate the inherent nonzero cumulants in the overall distribution of data we construct a rank-ordered map from the original dataset to a proxy dataset of the same length drawn from a Gaussian distribution [*Kennel and Isabelle*, 1992; *Schreiber and Schmitz*, 1996; *Deco and Schürmann*, 2000]. The distribution of the proxy dataset ensures that all cumulants of the distribution beyond second order should in principle vanish. However, the time-ordering of the data can still lead to nonzero cumulants, because the joint probability distribution of Dst(t+τ) and Dst(t) may be non-Gaussian even if the distribution of Dst is Gaussian. Moreover, it is simple to construct surrogate data from the Gaussianized data that shares the same autocorrelation by using the same power spectrum, but randomly shifting the phases of the Fourier coefficients. The surrogate data therefore has the same autocorrelation as the original data. Any deviation from the linear statistic is apparent from comparison with the surrogate data, and we interpret these deviations as evidence of nonlinear dependence because we have falsified the hypothesis that the data can be adequately described by linear statistics. That this method works is evident in Johnson and Wing [2005] where we compare analysis using mutual information (using the actual data) and higher order cumulants (using Gaussianized data) and find a very similar result when analyzing Kp data. We have added a paragraph in Section 3.1 to explain why we need to gaussianize the data.

4) If I have correctly understood the cumulant based method, the nonlinear cross-correlation quantity should provide an information of the overall (linear and nonlinear) correlation between VBs and Dst. Thus, how can the Authors state that peaks at 25, 50 and 90 hours are of an internal origin on the basis that they are not present in the auto- correlation of external drivers ? Furthermore, in doing their analysis the Authors have considered Dst records covering 27 years (1974-2001) without discriminating between single geomagnetic storms and multiple geomagnetic storms. So how they can assert that these secondary peaks (which is less

prominent) do not come from such multiple geomagnetic storms but reflects internal processes
? This conclusion seems to me not convincing. To convince the reader that there are secondary
peaks in the nonlinear cross-correlation that are of an internal origin, the Authors should make
the analysis on a subset of geomagnetic storms which are characterised by only a single negative-
peak in Dst.

We establish that there is a clear nonlinear response of Dst to VBs at lags = 3-10, 25, 50, and 90
hours.  However, in the self-significance of VBs, there are linear and nonlinear peaks at lags = 3-
12 hours.  We conclude that the peaks at lags = 25, 50, and 90 must be due to internal processes.

The argument is as follows:
Suppose that Dst is **completely** driven externally by VBs and VBs time series has multiple peaks
with 3 hours periodicity, then we would expect Dst to also have multiple peaks with 3 hours
periodicity and (VBs,Dst) significance to also have peaks with 3 hours periodicity.  However, if
(VBs, Dst) significance has peaks with 25 hours periodicity, then we can say that the origin of this
peak is not due to inherent nonlinearity in VBs.

If all the peaks in the Dst were externally driven, then in the case multiple storms, it would be
expected that the VBs would also have multiple peaks.  A peak or peaks in the self-significance of
VBs would also show up in the (VBs, Dst) significance.  On the other hand, if some of the peaks in
the Dst are not externally driven (internally driven), then there would be peaks in the (VBs, Dst)
significance that would not be present in the self significance of VBs.  The present study uses 27
years of data and should be seen as a statistical study.  Any rare or unusual features would appear
as small or insignificant peaks in the (VBs, Dst) significance because they have been "averaged"
out, but if the features are not rare, then the peaks would be significant.

On the other hand, we cannot entirely rule out other external drivers being responsible for the
evolution of Dst, but it is generally accepted that VBs is likely the most important driver for the
ring current decays in the recovery phase [e.g., Burton et al., 1975; O'Brien and McPherron, 2000;
McPherron and P'Brien, 2001; Weygand and McPherron, 2006], so at least we can conclude that
the nonlinearity seen in the response of Dst does not reflect the inherent nonlinearity of the
variable considered to be the most or one the most important driver, which is suggestive that
the nonlinear dependence identified in Dst is likely the result of magnetospheric processes.  We
have added a few sentences to clarify this point in Section 4 (summary).

5) Page 11. To my knowledge there should be also other processes/ mechanisms than ion
cyclotron waves -particle scattering that could be responsible for ring-current decay. For
instance, I remember that also ENA loss mechanisms could contribute to the decay of the ring
current. Perhaps, this could be considered in discussing this point.

Several studies found that the ring current decay has two stages due to different processes such
as convection of ions out of the front of the magnetopause, ENA charge exchange, and or
coulomb scattering [Hamilton et al., 1988, Ebihara et al., 1998, Kozyra et al., 2002, Macmahon
and Llop-Romero, 2011].  The ENA charge exchange can contribute to the ring current decay,

mainly in the late recovery phase or the second stage, which may begin about 1 day after the storm commencements [Kozyra et al., 2002]. The charge exchange can take several days to deplete the ring current to the baseline level [e.g., Smith et al., 1976]. The interplay of the multiple loss mechanisms may contribute to the multiple peaks in the (VBs,Dst) significance. We have added discussion on charge exchange and other loss mechanisms at the end of Section 3.1.

6) In the Transfer Entropy analysis section few details are given about the way Transfer Entropy and Mutual Information are computed. To my knowledge binning procedure and PDF computing method are critical issues in evaluating these quantities. I believe that more information should be provided to make the reader able to reproduce the results.

We used the same procedure as described in our previous work [Wing et al., 2016]. The number of bins ($n_b$) needs to be chosen properly, but fortunately, there are some guidelines that can be followed and usually there is a range of $n_b$ that would work. In general, we would like to maximize the amount of information. Having too few bins would lump too many points into the same bin, leading to loss of information. Conversely, having too many bins would leave many bins with 0 or a few number of points, which also leads to loss of information. *Sturges* [1926] proposes that for a normal distribution, optimal $n_b = \log_2(n) + 1$ and bin width ($w$) = range/$n_b$, where $n$ = number of points in the dataset, range = maximum value − minimum value of the points. In practice, there is usually a range of $n_b$ that would work. We have added a discussion on binning at the end of Section 2.2.

7) The result on the time delay (8-11 hr) between the information transfer from Vsw and Dst looks very long. The Earth's magnetosphere is expected to respond to solar wind changes on shorter timescale and this is also the case of ring-current. This is also corroborated by the capability of several Artificial Neural Network models of the Earth's magnetospheric response that consider a time delay of 1-2 hours as input variables for predicting Dst (see e.g. Wu and Lundstedt, JGR, 1997; Lundstedt et al., GRL, 2002; Pallocchia et al., Ann. Geophys., 2006). The Authors should motivate this result with more physical considerations.

Our result on the transfer of information from the VBs to the Dst with lag times of 8–11 is consistent with previous studies. For example, Borovsky et a. [1998] found that the solar wind takes 4 hr to reach the midnight region of the geosynchronous orbit and 15 hr to reach the noon region of the geosynchronous orbit. We have modified Section 3.2 to include this discussion.

Our data analysis aims to discover the dynamics of the magnetosphere, which may differ from that of a neural network model or any other models. We cannot say that we are familiar with the neural network models used for predicting Dst that are referenced by the reviewer. Hence, we would restrain from commenting on them. For example, we do not know if the modelers have considered inputting solar wind parameters in the last 11 hours and how their neural networks would respond to such a large input parameters. However, in a recent model for forecasting Dst, it has become evident that a long time history of solar wind parameters is

necessary. For instance, in [Lazzús, et al. (2017), Forecasting the Dst index using a swarm-optimized neural network, Space Weather] Dst forecast model, input parameters of the last 6 hours or more are used. It is not clear how the performance of this model compares with those of other models.

There are many paradigms of neural networks and each paradigm behaves differently. Our own experience working with neural networks is that the larger the number of input parameters, the larger the networks become and the harder the networks can generalize. On the other hand, there may be benefits from having large number of input parameters (or longer time history) as they may be needed to capture more fully the dynamics of the magnetosphere. So, we see that there is an inherent competition between having smaller input parameters (time history) vs. having larger input parameters (time history) in the neural networks that we work with.

8) Figure 1 is hardly readable. I suggest to expand the X-axis or to include a inset where the first part of X-axis is expanded.
We improved the readability of Figure 1. We think that it is not necessary to have an inset.

Minor points. Some references are missing (there are some question marks at page 8.
The missing references have been furbished. Thank you.